# Avascular Spaces of the Female Pelvis—Clinical Applications in Obstetrics and Gynecology

**DOI:** 10.3390/jcm9051460

**Published:** 2020-05-13

**Authors:** Stoyan Kostov, Stanislav Slavchev, Deyan Dzhenkov, Dimitar Mitev, Angel Yordanov

**Affiliations:** 1Department of Gynecology, Medical University Varna, 9000 Varna, Bulgaria; drstoqn.kostov@gmail.com (S.K.); st_slavchev@abv.bg (S.S.); 2Department of General and Clinical pathology, Forensic Medicine and Deontology, Medical University Varna, 9002 Varna, Bulgaria; dzhenkov@mail.bg; 3University hospital SBALAG “Maichin Dom”, Medical University Sofia, 1000 Sofia, Bulgaria; dr.dimitur.mitev@gmail.com; 4Department of Gynecologic Oncology, Medical University Pleven, 5800 Pleven, Bulgaria

**Keywords:** avascular spaces, surgery, applications in obstetrics, applications in gynecology

## Abstract

The term “spaces” refers to the areas delimited by at least two independent fasciae and filled with areolar connective tissue. However, there is discrepancy regarding the spaces and their limits between clinical anatomy and gynecologic surgery, as not every avascular space described in literature is delimited by at least two fasciae. Moreover, new spaces and surgical planes have been developed after the adoption of laparoscopy and nerve-sparing gynecological procedures. Avascular spaces are useful anatomical landmarks in retroperitoneal anatomic and pelvic surgery for both malignant and benign conditions. A noteworthy fact is that for various gynecological diseases, there are different approaches to the avascular spaces of the female pelvis. This is a significant difference, which is best demonstrated by dissection of these spaces for gynecological, urogynecological, and oncogynecological operations. Thorough knowledge regarding pelvic anatomy of these spaces is vital to minimize morbidity and mortality. In this article, we defined nine avascular female pelvic spaces—their boundaries, different approaches, attention during dissection, and applications in obstetrics and gynecology. We described the fourth space and separate the paravesical and pararectal space, as nerve-sparing gynecological procedures request a precise understanding of retroperitoneal spaces.

## 1. Introduction

The term “spaces” refers to the areas delimited by at least two independent fasciae and filled with areolar connective tissue. These spaces could be exposed by separating two independent fasciae along their cleavage plane [1,2]. However, there is discrepancy regarding the spaces and their limits between clinical anatomy and gynecologic surgery, as not every avascular space described in literature is delimited by at least two fasciae [2,3]. Ercoli et al. identified and defined some subdivisions of the main pelvic fasciae and spaces that are not officially recognized. Retroperitoneal spaces are useful anatomical landmarks in retroperitoneal anatomic and pelvic surgery for both malignant and benign conditions [2]. Moreover, new spaces and surgical planes have been developed after the adoption of laparoscopy and nerve-sparing procedures. The number of spaces varies from six to eight, as some authors separate the paravesical and the pararectal space into lateral and medial paravesical/pararectal spaces [1,2,3,4,5]. Three pairs of ligaments divide the retroperitoneal spaces [1,4]. These spaces are avascular and filled with fatty or loose areolar connective tissues [5]. Retroperitoneal spaces exist as the pelvic viscera are derived from different embryologic structures. Developing these spaces early during an operation exposes vital structures and avoids injuring the viscera, ureter, nerves, and blood vessels [1,4,5,6]. In order to perform safe dissection, the surgeon should be familiar with these spaces and their boundaries. Most of the spaces can be dissected either by minimally invasive (laparoscopic/robotic) or open (laparotomy) surgery and some of them with vaginal surgery. A noteworthy fact is that for various gynecological diseases, there are different approaches to the avascular spaces of the female pelvis [7]. This is a significant difference, which is best demonstrated by dissection of these spaces for gynecological, urogynecological, and oncogynecological operations. We define nine avascular female pelvic spaces. We describe the fourth space and separate the paravesical and pararectal space, as nerve-sparing gynecological procedures request a precise understanding of retroperitoneal spaces.

## 2. Avascular Spaces

The avascular spaces of the pelvis are named based on the location of the nearest organs (Figure 1).

These pelvic spaces are classified as follows [7]:Lateral spaces:
Paravesical (divided by umbilical artery into lateral and medial spaces)Pararectal (divided by ureter into lateral and medial spaces)The fourth space (Yabuki space)Median spaces:
Retropubic (Retzius) spaceVesicocervical/Vesicovaginal spacesRectovaginal spacePresacral or retrorectal space

## 3. Paravesical Space

Paravesical space boundaries: ventrally-superior pubic ramus, arcuate line of the os ilium; dorsally–cardinal ligament including parametrium (over the ureter) and paracervix (below the ureter), uterine artery/vein; medially-caudal portion of vesico-uterine ligament, bladder; laterally–obturator internus fascia/muscle, external iliac artery/vein. The paravesical space is covered by the peritoneum of the anterior leaf of broad ligament. Its floor is the ilio-coccygeus muscle and pubocervical fascia as it inserts into the arcus tendinous fascia pelvis [1,4,5,8,9,10,11]. Some authors consider the paravesical space as a lateral compartment of the Retzius space [8]. Paravesical space contains the umbilical artery, superior vesical artery, the obturator neurovascular bundle, lymphatic tissue, and accessory obturator vessels [10]. The obliterated umbilical artery and umbilical pre-vesical fascia divides this space into lateral paravesical space (LPS), and medial paravesical space (MPS), respectively [7,9] (Figure 2, Figure 3 and Figure 4).

The space within the LPS is known as obturator space. This space has the same boundaries as the paravesical space, except for a medially–superior vesical artery. The obturator space contains an obturator nerve/artery/vein, loose areolar, and lymphatic tissue [8]. The paravesical space is dissected by transecting the round ligament and cutting the anterior leaf of broad ligament ventrally and laterally to the obliterated umbilical artery [5]. Transecting the round ligament is not a necessary step. In our opinion, it is not necessary to transect the round ligament when performing laparoscopic pelvic lymphadenectomy previous to radical hysterectomy. The obliterated umbilical artery is identified and after ventral dissection, the paravesical space is opened to the level of levator ani muscle for the MPS and obturator nerve for LPS, respectively [5,7,9]. LPS and MPS can be dissected through laparotomy or laparoscopic/robotic surgery, for various gynecological conditions. From an oncogynecological point of view, the paravesical space is laterally open, medializing all anatomic structures, whereas in a benign gynaecological condition, the paravesical space is opened from a medial aspect and goes lateral to enter the space [11].

In urogynecology, LPS provides access for Burch colposuspension by avoiding the space of Retzius [5,12]. LPS is also developed for paravaginal defect repair without opening the MPS. After opening the LPS, the dissection is extended to the levator floor to expose the arcus tendineous fascia pelvis [12]. MPS is performed in cases of ureteric reanastomosis and paravaginal defect repair [9].

In the gynecology/oncogynecology, the paravesical space is opened in all cases of radical hysterectomy in order to obtain best identification of anatomical structures-parametrium/paracervix, vesicouterine ligament, ureter, bladder, pelvic vessels, nerves, and lymph nodes [5,7,9]. LPS is exposed for pelvic lymphadenectomy, whereas MPS gives access for management of bladder or ureteric endometriosis, or mobilization of the bladder during anterior exenteration. Paravesical space can be dissected through a vaginal approach for vaginal radical trachelectomy (Dargent’s operation), vaginal radical hysterectomy (Shauta’s operation), or through laparoscopically assisted radical vaginal hysterectomy. After opening the vesico-uterine space, the anterior vaginal wall is grasped at 9 o’clock and 11 o’clock positions and the right paravesical space is developed at the 10 o’clock position lateral to the vesico-uterine ligament. The same procedure is performed on the left sight (grasp the vagina at 3 o’clock and 1 o’clock positions; access to this space is located at the 2 o’clock position lateral to the vesico-uterine ligament [13]. Although rare, the paravesical space is developed through an extraperitoneal approach. This approach is carry out to perform extraperitoneal pelvic lymphadenectomy in oncogynecological cases. After a 12-cm midline incision in the lower abdomen, the space between the rectus abdominis sheath and the parietal peritoneum is developed toward the left inguinal region in order to identify the external iliac artery and vein. The paravesical space is exposed after separation of the peritoneal sac containing intraperitoneal organs from the external iliac artery/vein [14,15]. In 1960, Mitra first described a combination of extraperitoneal bilateral pelvic lymphadenectomy with vaginal radical hysterectomy for carcinoma of the cervix [16]. The first step of the operation was an extra peritoneal pelvic lymph node dissection through bilateral suprainguinal incisions. In addition, Mitra ligated ovarian/uterine vessels and partial dissected ureteric canals/paravesical/pararectal spaces [16]. At times, after neoadjuvant pelvic radiation therapy, it is difficult to develop a pararectal space. In these cases, the paravesical space should be opened. After transecting, the cardinal ligament in the pararectal space is opened inferiorly with the ureter identified medially and the iliac veins visualized laterally (17).

Paravesical space is dissected during detection of the sentinel lymph node due to cervical or endometrial cancer [17].

In obstetrics, a paravesical space is developed during cesarean hysterectomy for placenta percreta in order to clip uterine arteries and identify the ureter [18]. Vesicouterine and paravesical spaces are exposed for minimally invasive (laparoscopic/robotic) or open (laparotomy) cerclage. Whittle reported 65 patients who underwent laparoscopic cervico-isthmic cerclage. Whittle’s surgical technique includes development of paravesical and vesico-uterine spaces and creation of broad ligament peritoneal windows due to identification of the uterine vessels at the cervico-isthmic junction [19].

Attention during dissection of PVS: The definition of Corona mortis (CMOR) is heterogeneous. Some authors define CMOR and accessory obturator vessels as the same structures, whereas others define CMOR and accessory obturator vessels as different structures [4,20]. CMOR is a potential vascular anastomosis between the external iliac/inferior epigastric and obturator vessels. In our opinion, accessory obturator vessels differ from CMOR as not all of these vessels connect the obturator vessels to the external iliac system. Moreover, accessory obturator arteries originate from the external iliac/inferior epigastric arteries and pierce the obturator membrane, without participating in anastomosis. Accessory obturator veins drain in the external iliac/interior epigastric veins [20]. CMOR varies extensively in terms of whether it is arterial, venous, or both. A broad study by Sanna et al. included 2184 hemi-pelvises. The overall prevalence of corona mortis was 49.3%. The prevalence of venous CMOR was greater than an arterial CMOR (41.7% vs. 17.0%). CMOR runs dorsally to the superior pubic ramus over the obturator fossa. Accurate anatomical knowledge of the CMOR and accessory obturator vessels is vital in order to perform safe oncological pelvic dissection or Burch colposuspension [5,10,20] (Figure 5).

## 4. Pararectal Space 

Pararectal space boundaries: ventrally–cardinal ligament, including the parametrium (over the ureter) and paracervix (below the ureter); dorsally–presacral fascia, sacrum; laterally–internal iliac artery; medially–rectum; cranially-peritoneum of the posterior leaf of the broad ligament; caudally-levator ani muscle [1,4,5,7,9,10,11]. There are two different approaches to the pararectal space–oncological (lateral) and endometriotic (medial), as reported by Puntambekar. In the lateral approach, the peritoneum is incised lateral to the infundibulopelvic ligament, whereas in the medial approach, the incision is medial to the infundibulopelvic ligament [7]. The first structure visualized through the medial approach is the ureter. After a slight dissection, the ureter divides the pararectal space (PRS) into medial (Okabayashi’s space) and lateral (Latzko’s space) pararectal spaces [5,7]. Lemos et al. reported that the hypogastric nerve is the structure dividing the PRS into lateral and medial [21].We agree with this proposal, as the ureter and the hypogastric nerve (HN) are in the same connective tissue plane (mesoureter), and the HN is localized 2–3 cm dorsally to the ureter [1,22] (Figure 6, Figure 7 and Figure 8).

Latzko’s space boundaries: ventrally-cardinal ligament; dorsally- presacral fascia, ventrolateral aspect of the sacrum, laterally-internal iliac artery (hypogastric artery); medially–ureter, mesoureter.

The incision is made in the peritoneum at the level of the pelvic brim after transperioneal identification of the ureter, remaining parallel and lateral to the ureter. Development of this space starts at the level of iliac vessels’ bifurcation. Latzko’s space is exposed through craniocaudal and dorsoventral dissection between the internal iliac artery and the ureter [1,4,5,7,9,23].

In gynecology/oncogynecology, Latzko’s space is developed during pelvic lymphadenectomy or in cases requiring access to the pelvic splanchnic nerves, inferior hypogastric plexus (pelvic plexus). The dissection of Latzko’s space exposes the uterine artery originating from the internal iliac artery. The procedure is performed for various cases—radical hysterectomy, adhesions from previous pelvic surgeries, large myomatous uterus, temporary ligation with clips for abdominal, laparoscopic hysterectomy or myomectomy [7,9]. Although it is not necessary, the approach for this procedure starts with transection of the round ligament. The posterior leaf of the broad ligament is incised laterally to the infundibulopelvic ligament. After identifying the ureter and internal iliac artery, the dissection proceeds caudally between these two landmarks and leads to the uterine artery originating from the hypogastric artery [24]. 

In obstetrics and gynecology, Latzko’s space is developed during internal iliac artery ligation to control pelvic hemorrhage. The peritoneum overlying the psoas muscle is incised 8-9 cm parallel to the external iliac artery and lateral in the line with the ureter. Further dissection exposes common iliac artery, bifurcation of the internal iliac artery, and the internal iliac vein. The Latzko’s space is then opened next for identification and medialization of the ureter, along with lateralization of the internal iliac artery [25]. A right angle forceps is passed between the internal iliac artery and vein distal to the posterior division of the artery (in a majority of cases superior gluteal, lateral sacral, and lumbosacral arteries), which supply the buttocks and gluteal muscles [1]. The artery is ligated with a non-absorbable suture. Complications such as external or common iliac artery ligation, internal iliac vein or ureter injury, buttock claudication or pelvic ischemia might occur [1,17,25].

Okabayashi’s space boundaries: ventrally-cardinal ligament; dorsally–presacral fascia, sacrum; laterally–ureter, mesoureter; medially-rectum. The authors define different medial boundaries of Okabayashi’s space. For some authors, the medial limit of Okabayashi’s space is the rectum, whereas for others, it is the uterosacral ligaments [2,5,9,11,22,23,26]. In our opinion, the medial limit of Okabayashi’s space is the rectum. However, as there is some discrepancy between surgeons, further studies are needed to clarify the medial limit of Okabayashi’s space. By incising and opening the space between the posterior leaf of the broad ligament and the ureter, Okabayashi’s space is developed. This step allows lateralization of the ureter, mesoureter, HN, and gives access to the uterosacral ligament [5,9,27]. Two anatomic structure transversely cross this space—the lateral ligament of the rectum (LLR) and middle rectal artery (MRA). LLR is formed by the MRA/vein and surrounding tissue. LLR and MRA are stretched between the hypogastric vessels and the rectum. Although LLR and MRA are absent sometimes, they are of significant importance during nerve-sparing procedures. LLR and MRA are anatomic landmarks—after slight dissection caudally to LLR and MRA, the pelvic plexus is visualized. LLR and MRA runs close to the pelvic splanchnic nerves and transverse the pelvic plexus [1,2,11,28].

In gynecology/oncogynecology, Okabayashi’s space is exposed during nerve-sparing radical hysterectomy or in cases with deep endometriosis in order to preserve nerves. Dissection of Okabayashi’s space is essential for identification and dissection of hypogastric nerves [23]. Fujii reported that during nerve-sparing radical hysterectomy, developing Okabayashi’s space is not required, as the development of Latzko’s space is enough [22]. 

We dissect the Okabayashi’s space during laparoscopic uterosacral nerve ablation (LUNA). Identifying the ureter avoids ureteral injury during uterosacral ligaments transection. 

Okabayashi’s space is also dissected for ureter mobilization, bowel management, or major ureter surgery in deep endometriosis [9,23].

PRS can be dissected when it is approached by the vaginal route for sacrospinous ligament fixation procedure, vaginal radical trachelectomy (Dargent’s operation), vaginal radical hysterectomy (Shauta’s operation), or laparoscopically assisted radical vaginal hysterectomy. The left PRS is dissected by placing two Kohler forceps at 3 o’clock and 5 o’clock positions. After outward traction to these forceps, a depression between them (4 o’clock) becomes visible. The left PRS is opened downward and outward with a scissors [17,29].

As the paravesical space, the PRS can be opened by the extraperitoneal approach, which has been described above.

The pararectal space is dissected during sentinel lymph node biopsy. 

Attention during dissection of PRS: During dissection, making a deep narrow hole with a bottom that cannot be exposed is not recommended, as the deep vein might be lacerated. Dissection of the caudal limit of PRS must be made cautiously to avoid damage to the lateral sacral and hemorrhoidal vessels. Developing of this space proceeds after identification of the ureter [17].

## 5. Yabuki Space

The Yabuki space, also called the fourth place, was first described in 2000 by Yoshihiko Yabuki. It is located between the cranial portion of the vesicouterine ligament and the ureter. The Yabuki space is dissected during nerve-sparing surgery as it contains the pelvic splanchnic nerves on the way for bladder innervation (Figure 9).

The Yabuki space is exposed after dissection of the cranial portion of the vesico-uterine ligament. With the ureter emerging as a target, the fourth space is exposed by dissection and coagulation. Dissection proceeds to the vesicoureteral junction [23,30]. Later in 2005, Yabuki reported the same space as Okabayashi’s paravaginal space [3]. Liang et al. concluded that the Yabuki space should be located between the lateral side of the vagina and the caudal portion of the vesico-cervical ligament [30]. Nerve-sparing techniques in oncologic pelvic procedures are based on anatomical landmarks—four spaces (medial paravesical, Okabayashi, Latzko, Yabuki spaces) and four structures (the ureter, LLR, MRA, and the deep uterine vein) [1,21].

## 6. Retropubic (Retzius) Space

Retzius space boundaries: ventrally—pubic symphysis; dorsally—parietal peritoneum, bladder; cranially—transversals fascia; caudally—the anterior aspects of the urethra, adjacent pubocervical fascia, and bladder neck; laterally—the arcus tendinous fasciae pelvis, which lies on the inner surface of the obturator internus and the pubococcygeal and, puborectalis muscles [1,8,11,17,31] (Figure 10).

The Retzius space is a potential extraperitoneal space. There are discrepancies between paravesical and retropubic space limits in classic anatomy and modern gynecologic surgery. In the literature, there is no clear distinction between the Retzius space boundaries and the paravesical space, as the paravesical space lies on either side of the Retzius space. Lateral limit of the retropubic space are the paravesical spaces, as reported by some authors, the line of demarcation being the obliterated umbilical artery. Other authors divide the paravesical space into LPS and MDS [2,5,6,7]. In our opinion, laparoscopic nerve-sparing operations for deep endometriosis and gynecological cancer provide new vision of the spaces and it is more appropriate to divide the paravesical spaces.

Moreover, there are multiple fascial layers present in the Retzius space, so it is not a single anatomic entity. In his research study, Ansari observed multiple potential spaces in the retropubic region. He concluded that the Retzius space needs to be re-evaluated and re-defined for anatomist and practicing surgeons [32]. We totally agree with Ansari, as in the retropubic region, there are fascial layers and spaces, which have not been mentioned in anatomical textbooks or during surgical dissection. Fascial layers, such as the umbilical vesical fascia and the umbilical prevesical fascia, have not been recognized in most of the laparoscopic intraperitoneal approaches. This is not correct, as the umbilical vesical fascia (not the transversals fascia) is the first visceral layer to be dissected with the intraperitoneal approach. We described the classical cavum Retzii boundaries. However, further anatomical studies are needed for better clarification.

The Retzius space has great clinical significance in urogynecology and is more often reached by laparoscopic approach.

*In urogynecology,* Retzius space is very useful mainly for stress urinary incontinence procedures, i.e., Burch colposuspension, retropubic TVT, MESH removals as well as anterior vaginal compartment repairs [31]. Nowadays, Marshall-Marchetti-Krantz procedure for stress urinary incontinence is rarely performed.

In oncogynecology, we enter this space during anterior exenteration and pelvic anterior peritonectomy.

In gynecology, it is developed for bladder endometriosis [23].

The urachus (median umbilical ligament) lies on the inner aspect of the anterior abdominal wall dorsal to the fascial layers (umbilical vesical fascia, umbilical prevesical fascia, transversalis fascia). The urachus and both medial umbilical ligaments are used as landmarks, just above the bladder dome, as the site for the peritoneal incision [31,32]. After cutting the median umbilical ligament, dissection proceeds in the ventro-lateral direction up to the retropubic space boundaries.

Attention during dissection: The presence of several blood vessels is important and surgeons have to be familiar with their location. There is a large plexus of veins, known as the veins of Santorini, which is identified within the paravaginal tissue. The veins of Santorini drains into the internal iliac vein. The dorsal vein of the clitoris runs caudally to pubic symphysis and drains into the plexus of Santorini [6,31]. If the surgeon loses the correct avascular plane, blood or bladder injury might occur.

## 7. Vesicocervical/Vesicovaginal Space

The vesicovaginal space is the inferior extension of the vesicocervical space and they are in the same longitudinal axis. The area is also recognized as the anterior cul-de-sac [5]. 

Vesicocervical space boundaries: ventrally—bladder; dorsally—pubocervical fascia, cervix (upper part); laterally—cranial portion of vesicouterine ligament; cranially—peritoneal reflection between the dome of the bladder and the low uterine segment in the region of the anterior cul-de sac. 

Vesicovaginal space boundaries: ventrally—trigone of bladder; dorsally—pubocervical fascia, vagina (inferior part); laterally—cranial portion of vesicouterine ligament; cranially—peritoneal reflection between the dome of the bladder and the low uterine segment; caudally—junction of the proximal and middle thirds of the urethra. Below the caudal limit of vesicovaginal space, the anterior wall of the vagina and the posterior wall of the urethra fuse [5,10,11,32,33,34] (Figure 11).

Multiple surgical approaches may be applied to open vesicocervical/vesicovaginal spaces. 

In abdominal, laparoscopic, and robotic approaches, the uterovesical peritoneum is incised and dissected centrally on the pubovesical fascia of the cervix and upper vagina [8]. 

These spaces can also be entered surgically though the vaginal route. The anterior vaginal wall is incised transversely and the dissection remains centrally on the cervix to incise the supravesical septum. The vesicouterine peritoneum is exposed and cut in order to enter the anterior cul-de-sac [8].

In urogynecology, the vesicovaginal space is exposed for vesicovaginal, ureterovaginal fistula repair, urinary stress incontinence procedures, transvaginal cystocele operations (anterior colporrhaphy), and during laparoscopic sacrocolpopexy or hysterocolpopexy [4,33,34,35,36].

In gynecology/oncogynecology, vesicocervical/vesicovaginal spaces are developed during vaginal, abdominal, laparoscopic, and robotic hysterectomy. The dissection of the vesicovaginal space is performed until its caudal limit during radical hysterectomy in order to remove the upper one-third, half, or upper three quarters of the vagina. Vesicovaginal and paravesical spaces are developed for nerve-sparing radical hysterectomy or management of deep endometriosis. Dissection exposes the cranial and caudal portion of the vesico-uterine ligament, which allows meticulous dissection of vessels and nerves, described precisely by Fujii [22].

Dissection of the vesicovaginal space is required for vaginal cuff resection. It is performed for vaginal vault endometriosis or vaginal cuff recurrence after oncogynecological procedures [36].

In obstetrics, the vesicouterine space is approached for cesarean section, cesarean hysterectomy, or minimally invasive (laparoscopic/robotic), open (laparotomy) cerclage. The development of the vesicovaginal space with exposure of the lower segment is required in cases of cesarean section scar ectopic pregnancy [36].

Attention during dissection: It is important to stay above the pubovesical fascia in order to avoid bleeding and losing the right plane. The dictum followed here is “fat belongs to the bladder” [8,23]. The cranial vesicouterine ligament contains: uterine artery, superficial uterine vein, ureter branch of the uterine artery, superior vesical vein that drains into the superficial uterine vein, and cervicovesical vessels [22]. Nakamura reported that cervicovesical vessels are a branch of the superior vesical artery [37]. The ureter runs between the cranial and caudal portion of the vesicouterine ligament. Dissection of the vesicouterine/vesicovaginal space should be performed medially as lateral dissection will lead to vessels or ureteric injury. In laparoscopic procedures, entry to the vesico–uterine/vaginal space is achieved by pulling up the bladder. Only pulling up the peritoneum increases risks of bladder injury during dissection (Figure 12).

## 8. Rectovaginal Space

Rectovaginal space boundaries: ventrally—posterior vaginal wall; dorsally—anterior rectal wall; laterally—uterosacral ligaments (cranial), rectovaginal ligament (caudal); cranially—peritoneal reflections of the pouch of Douglas; caudally—levator ani muscle [1,5,7,11] (Figure 13).

This area is also recognized as the posterior cul-de-sac [17]. There are two different approaches to the rectovaginal space (RVS)—medial and lateral. Through the medial approach, the rectouterine pouch is cut and the Denonvilliers’ fascia is exposed. Denonvilliers’ fascia is more prominent in younger patients and corresponds to rectovaginal septum in females.

There are many controversies in the literature regarding this fascia, as the appearance of the fascia during operations varies considerably [38]. It was first described in men by the French anatomist Denonvilliers, who named this layer the “rectovesical septum”. Some authors have described its existence in women, whereas others fail to find its presence [39,40]. In his case study, Zhai concluded that the rectovaginal septum was composed of two layers—the anterior layer (Denonvilliers’ fascia) and the posterior layer (fascia propria of the rectum) [40]. According to Puntambekar, there are two layers of Denonvilliers’ fascia—one covers the rectum and the other the posterior vaginal wall [23]. 

However, there are avascular planes between these two fascial layers and the medial dissection should proceed between them in a cranial caudal direction until the caudal limit of the rectovaginal space is reached.

If the dissection plane is loose, vaginal bleeding or rectum injury might occur [23]. Ceccaroni described a different lateral approach for treatment of deep endometriosis. After opening both pararectal spaces, the development of the rectovaginal septum proceeds in latero-medial, cranio-caudad, and dorso-ventral directions. The lateral approach allows to surround the disease from the back after identifying the ureter and hypogastric nerves [9,27].

The vaginal route enters this space after vertical or transverse posterior vaginal incision at the junction of the lower third or middle third of the posterior vagina [8]. The RVS space is reached through a vaginal approach for rectocele repair [35].

In urogynecology, the rectovaginal space is developed during laparoscopic sacrocolpopexy or uterosacral ligament suspension for the treatment of vaginal vault prolapse [41].

In gynecology/oncogynecology, the rectovaginal space is exposed for radical hysterectomy, pelvic adhesions, treatment of deep endometriosis, or rectovaginal fistula repair.

Attention during dissection: The main rule is “fat belongs to the rectum”. For safe dissection, the goal is to stay between two fascial layers to avoid vessel or rectal injury. Vessel injury can occur from the middle rectal artery/vein, vaginal veins, and presacral veins if rules of dissection are not followed [23].

## 9. Presacral/Retrorectal Space

Presacral/retrorectal space boundaries: ventrally—mesorectal fascia/rectum; dorsally—longitudinal anterior vertebral ligament, sacral promontory, anterior aspect of the sacrum; laterally—right (right common iliac artery/right ureter), left (left common iliac vein/left ureter), hypogastric fascia, which is formed by the medial fibers of the uterosacral ligaments; cranially—peritoneal reflection of the rectosigmoid colon; caudally—pelvic floor [8,10,11,21,42] (Figure 14).

This space is divided by the Waldeyer’s fascia into inferior and superior retrorectal space [43]. There are three fascial layers of the presacral area—presacral fascia, proper rectal fascia (mesorectal fascia), and Waldeyer’s fascia. These fascial layers determine different planes and approaches. The term “Waldeyer’s fascia” causes significant confusion in gynecology and modern rectal surgery [11,43,44]. Wilhelm Waldeyer described the floor of the retrorectal space as the fascia, arising from the distal, posterior fusion of the visceral and parietal pelvic fascias found above the anococcygeal ligament [43,45,46]. Some authors concluded that the term “Waldeyer’s fascia” likely refers to the rectosacral fascia, whereas others suggested that rectosacral and Waldeyer fascia are two distinct anatomical structures [9,42,43,44,46]. Jin described Waldeyer’s fascia as fascia, which is fused posteriorly with the presacral fascia and anteriorly with the posterior leaf of the rectal parietal fascia [43]. Patel described this fascia as a caudal limit of presacral/retrorectal space [42]. Garcia-Armengol concluded that the terms Waldeyer’s fascia and retrosascral fascia are different structures owing to their different topographical sites [46]. We perceive that the majority of authors describe Waldeyer’s fascia as rectosacral fascia, which originates at the fourth portion of the sacrum, runs caudally, and fuses with the posterior leaf of fascia propria of the rectum. We conclude that the most important aspect is to identify this fascial layer (either Waldeyer’s fascia or retrorectal fascia) and failure to recognize and divide it can cause rectal perforation or severe presacral hemorrhage. Division of this fascia can provide access to the inferior portion of the retrorectal space, resulting in successful mobilization of the rectum during operations for bowel endometriosis or ovarian cancer infiltrating the rectal wall [43,44].

In oncogynecology, the inter-fascial approach distinguishing between mesorectal fascia and presacral fascia is known as the holy plane of dissection. It is used for total mesorectal excision (TME). This procedure is performed for ovarian cancer infiltrating the rectal wall, as the pattern of lymphatic spread may be similar to that of primary rectal carcinoma [47]. The inter-fascial approach is sometimes used for laparoscopic nerve-sparing complete excision of rectal endometriosis [9,27,48].

In gynecology, for patients with a benign disease such as endometriosis, the trans-mesorectal approach is found to be superior to the inter-fascial. The trans-mesorectal approach preserves the mesorectum and severe complications can be avoided. Using histologic findings, Mangler et al. showed in their study that the mesorectum is always free of endometriotic lesions, and thus preserving this region is justified. Moreover, studies showed that mesorectum preservation reduces nerves and ischemic vascular complications [11,49,50]. 

The presacral space is dissected by pulling the rectosigmoid to the right side of the pelvis and incising the peritoneum vertically to the left sigmoid peritoneal attachment to the posterior pelvis. The dissection starts at the aortic bifurcation and proceeds caudally. For entry into the presacral space, the left common iliac vein is the structure, which is at high risk of injury. Therefore, it must be identified (35). Another approach is at the level of the promontory. There are two layers of connective tissue—peritoneum along with its accompanying areolar tissue and presacral fascia containing the superior hypogastric plexus and nerves. The incision of the peritoneum starts from the promontory along the axis of the right common iliac artery. The underlying presacral fascia is dissected along the medial border of the right common iliac artery in the cranio-caudal and latero-lateral directions. The prelumbar space is opened to identify the longitudinal anterior vertebral ligament. Presacral fascia containing middle sacral vessels, superior hypogastric plexus, and hypogastric nerves is preserved [33].

In urogynecology, the prelumbar space is developed for sacrocolpopexy [33].

In gynecology/oncogynecology, the prelumbar space is exposed during presacral neurectomy or at the beginning of para-aortic lymphadenectomy [35,36].

Attention during dissection: At the beginning of dissection, common iliac vessels, inferior mesenteric artery, and ureters are at high risk of injury. The right plane between fascial layers has to be followed in order to avoid middle sacral vessels or nerve injury. A hemorrhage from the middle sacral vessels is difficult to control, as these vessels are difficult to clamp or suture. The superior hypogastric plexus and hypogastric nerves are very fine nerves, hardly visible even by laparoscopy. The operator should excise tissues when all the bordering anatomic structures are visible [8,35].

We summarized the application of avascular spaces in Table 1.

## 10. Conclusions

Avascular spaces of the female pelvis have many applications not only in gynecology, but also in obstetrics. Significance of all retroperitoneal female spaces in obstetrics is summarized and mentioned for the first time. These spaces and associated structures should be fully understood by the surgeon. Thorough knowledge of the pelvic anatomy of these spaces is important in retroperitoneal pelvic surgery. However, there are discrepancies regarding their boundaries. Therefore, more studies are needed for complete description of the avascular spaces of the female pelvis, which may be useful for both teaching and clinical purposes [2,3,36].

## Figures and Tables

**Figure 1 jcm-09-01460-f001:**
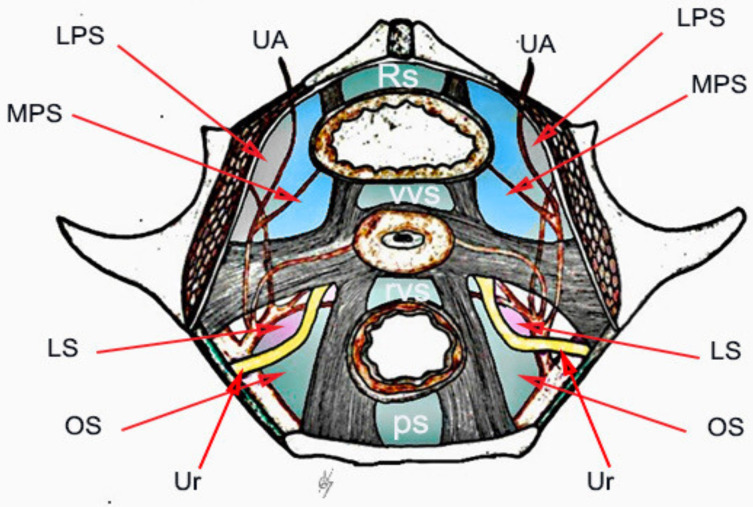
Avascular spaces of the female pelvis. UA–obliterated umbilical artery; Ur–ureter; LPS–lateral paravesical space; MPS–medial paravesical space; LS–Latzko’s space; OS–Okabayashi’s space; RS–Retzius space; VVS–vesicovaginal space; RVS–rectovaginal space; PS–presacral space.

**Figure 2 jcm-09-01460-f002:**
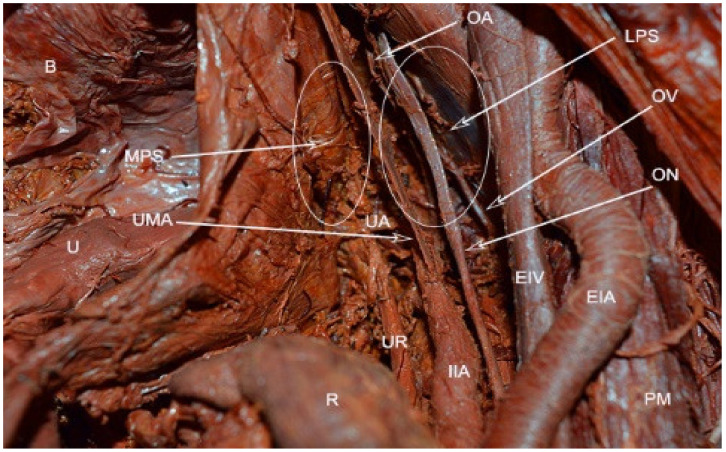
Paravesical space—embalmed cadaver (right side of the pelvis). The anterior leaf of the broad ligament is dissected and stretched laterally. MPS—medial paravesical space; LPS—lateral paravesical space; OA—obturator artery; OV—obturator vein; ON—obturator nerve; UA—uterine artery; UMA—obliterated umbilical artery; UR—ureter; EIV—external iliac vein; EIA—external iliac artery; PM—psoas muscle; IIA—internal iliac artery; R—rectum; U—uterus; B—bladder.

**Figure 3 jcm-09-01460-f003:**
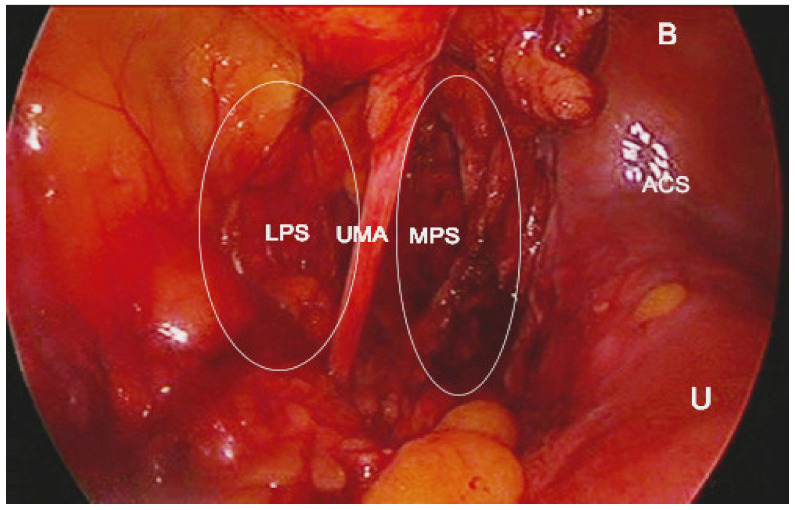
Paravesical space developed by laparoscopy (left side of the pelvis). MPS—medial paravesical space; LPS—lateral paravesical space; UMA—obliterated umbilical artery; U—uterus; B—bladder; ACS—anterior cul-de sac.

**Figure 4 jcm-09-01460-f004:**
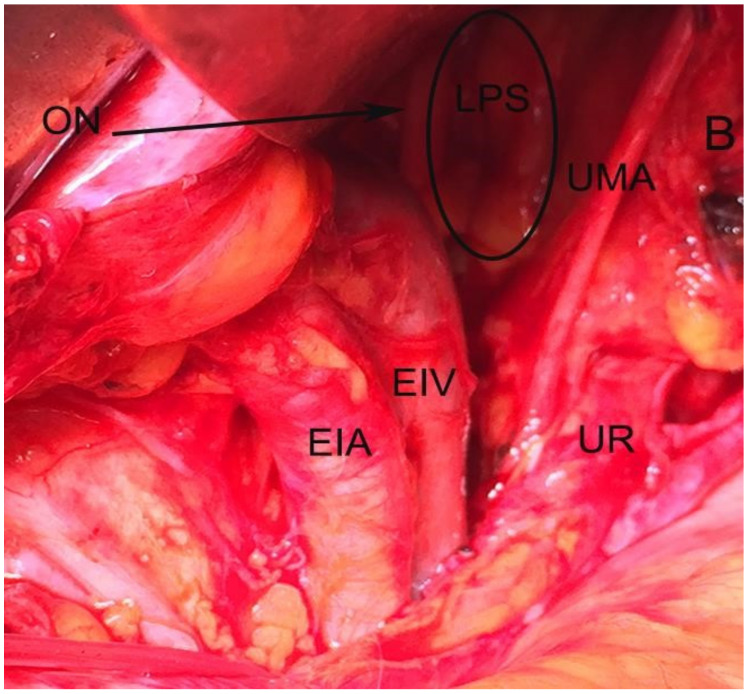
Lateral paravesical space dissected by open surgery (left side of the pelvis). LPS—lateral paravesical space; ON—obturator nerve; UMA—obliterated umbilical artery; UR—ureter; EIV— external iliac vein; EIA—external iliac artery; B—bladder.

**Figure 5 jcm-09-01460-f005:**
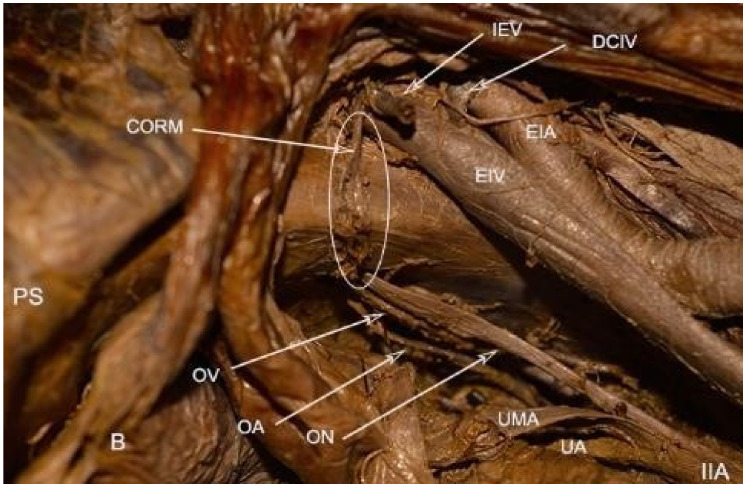
Corona mortis—embalmed cadaver (right side of the pelvis). Rare case of anastomosis between obturator vein and inferior epigastric vein. CORM—corona mortis; PS—pubic symphysis; B—bladder; OV—obturator vein; OA—obturator artery; ON—obturator nerve; IEV—inferior epigastric vein; DCIV—deep circumflex iliac vein; UMA—obliterated umbilical artery; UA—uterine artery; IIA—internal iliac artery; EIA—external iliac artery; EIV—external iliac vein.

**Figure 6 jcm-09-01460-f006:**
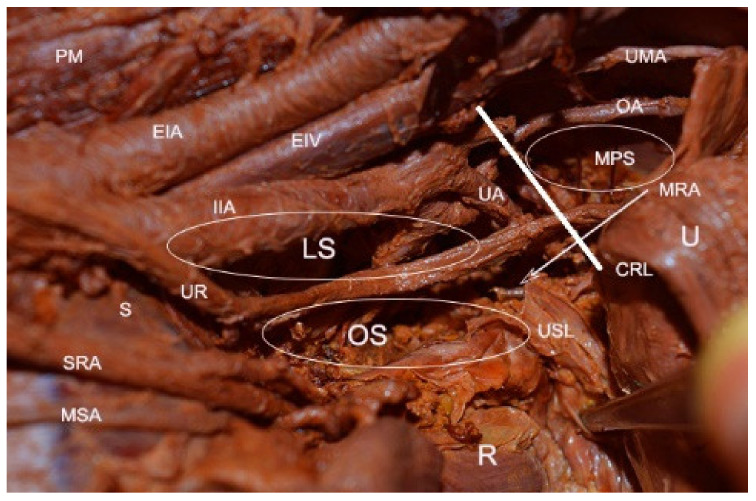
Pararectal space-embalmed cadaver (left side of the pelvis). LS—Latzko’s’ space; OS—Okabayashi’s space; PM—psoas muscle; EIA—external iliac artery; EIV–external iliac vein; IIA—internal iliac artery; UR—ureter; S—sacrum; SRA—superior rectal artery; MSA—middle rectal artery; R—rectum; U—uterus; CRL—cardinal ligament (white line); USL—uterosacral ligament; MRA—middle rectal artery (incised); MPS—medial paravesical space; OA—obturator artery; UMA—obliterated umbilical artery; UA—uterine artery.

**Figure 7 jcm-09-01460-f007:**
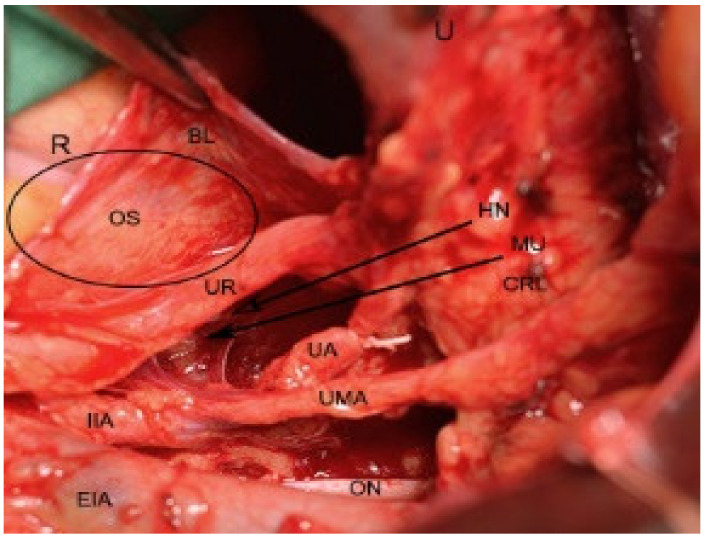
Development of Okabayashi’s space during open surgery (right side of the pelvis). The ureter and mesoureter are separated from the posterior leaf of broad ligament. R—rectum; BL—posterior leaf of broad ligament; U—uterus; HN—right hypogastric nerve; OS—Okabayashi’s space development; MU—mesoureter; CRL—cardinal ligament; ON—obturator nerve; UMA—umbilical artery; UA—uterine artery; UR—ureter; EIA—external iliac artery; IIA—internal iliac artery.

**Figure 8 jcm-09-01460-f008:**
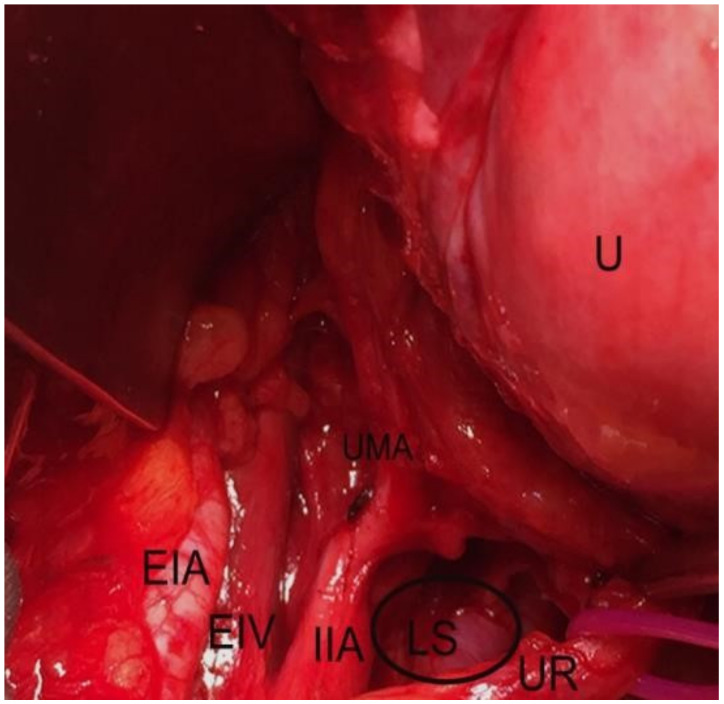
Latzko’s space dissected by open surgery (left side of the pelvis). U—uterus; UMA— umbilical artery; IIA—internal iliac artery; EIV—external iliac artery; EIV—external iliac vein, UR— ureter; LS—Latzko’s space.

**Figure 9 jcm-09-01460-f009:**
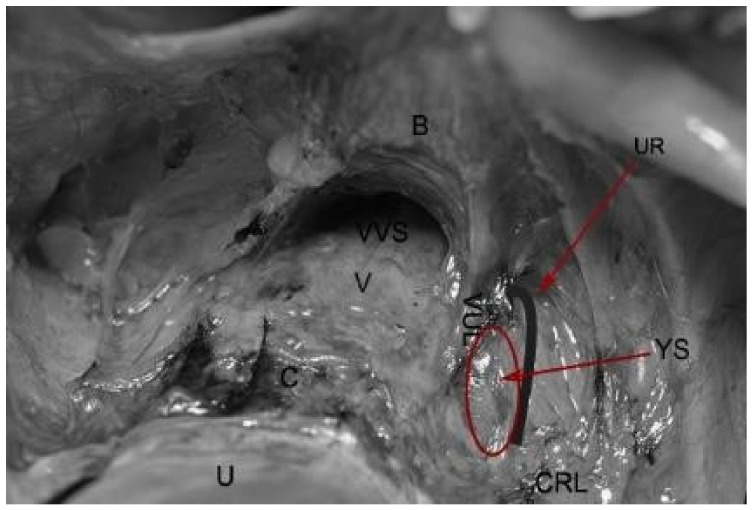
Yabuki space. U—uterus, C—cervix, V—vagina; VVS—vesicovaginal space; B—bladder; VUL—cranial portion of vesico-uterine ligament; UR—ureter; CRL—cardinal ligament; YS—Yabuki space.

**Figure 10 jcm-09-01460-f010:**
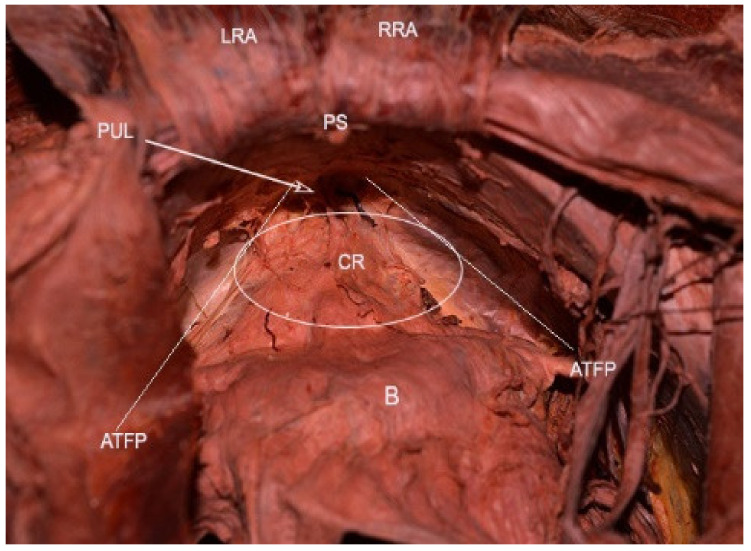
Retzius space (embalmed cadaver). ATFP—arcus tendinous fascia pelvis; PUL—pubourethral ligament; LRA—left rectus abdominis; RRA—right rectus abdominis; PS—pubic symphysis; B—bladder; CR—cavum Retzii..

**Figure 11 jcm-09-01460-f011:**
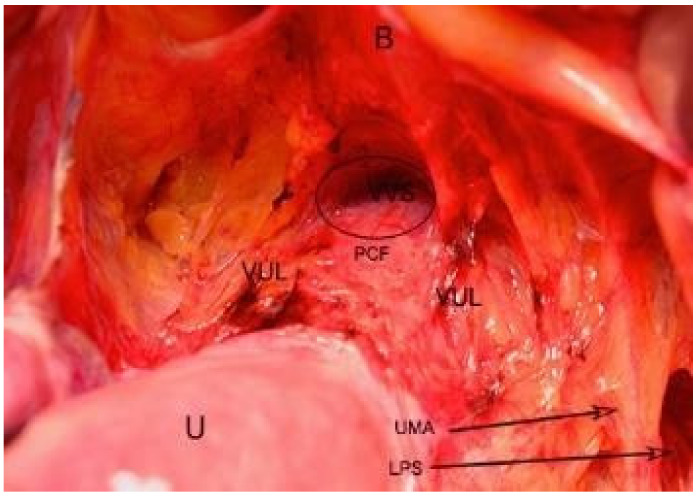
Vesicovaginal space dissected by open surgery. B—bladder; U—uterus; UMA—umbilical artery; LPS—lateral paravesical space; VUL—cranial portion of the vesico-uterine ligament; PCF— pubocervical fascia; VVS—vesicovaginal space.

**Figure 12 jcm-09-01460-f012:**
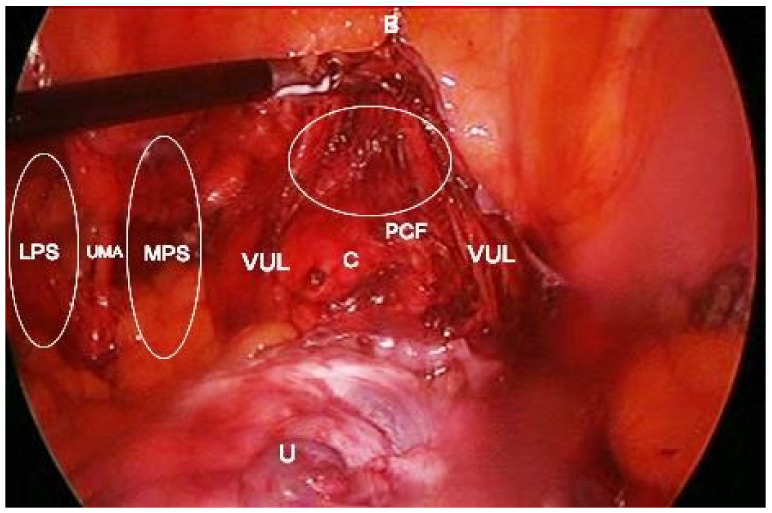
Development of vesicovaginal space during laparoscopic total hysterectomy shown in empty ellipse. The bladder is pulled up with a grasper. B—bladder; U—uterus; MPS—medial paravesical space; LPS—lateral paravesical space; C—cervix; UMA—umbilical artery; VUL—cranial portion of the vesico-uterine ligament; PCF—pubocervical fascia.

**Figure 13 jcm-09-01460-f013:**
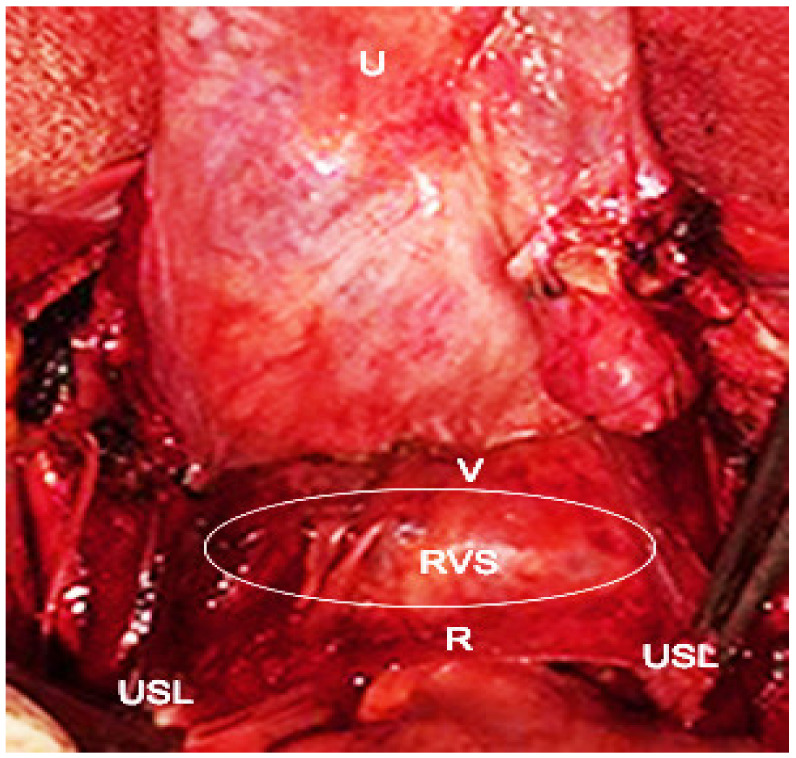
Rectovaginal space. U–uterus; USL–uterosacral ligaments are cut and pick up with tweezers; R–anterior rectal wall; V–posterior vaginal wall; RVS–rectovaginal space.

**Figure 14 jcm-09-01460-f014:**
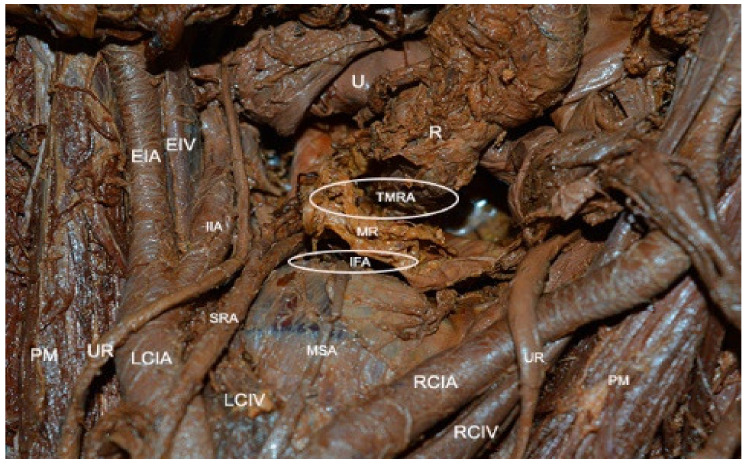
Presacral space (embalmed cadaver). PM—psoas muscle; LCIV—left common iliac vein; LCIA—left common iliac artery; UR—ureter; MSA—middle sacral artery; PR—promontory; RCIV—right common iliac vein; SRA—superior rectal artery; RCIA—right common iliac artery; EIA—external iliac vein; IIA—internal iliac vein; MR—mesorectum; R—rectum; U—uterus; TMRA—trans-mesorectal approach; IFA—inter-fascial approach.

**Table 1 jcm-09-01460-t001:** Avascular spaces—applications and attention during dissection.

Avascular Spaces	Application of Avascular Spaces			Attention during Dissection
	Oncogynecology/Gynecology	Urogynecology	Obstetrics	
**LPS**	Pelvic lymphadenectomy, RH, RVH, SLNB	Burch colposuspension, paravaginal repair	Cesarean hysterectomy, LA/A cerclage	CORM
**MPS**	Anterior exenteration, RH, DE treatment, RVH, SLNB	Ureteric reanastomosis, paravaginal repair	Cesarean hysterectomy, LA/A cerclage	CORM
**Latzko’s’ space**	Pelvic lymphadenectomy, RH, uterine artery ligation, nerve-sparing procedures, RVH, sentinel lymph node biopsy, SLNB	Ureter surgery for DIE or GC	Internal iliac artery ligation	Lateral sacral/hemorrhoidal vessels, pelvic splanchnic nerves
**Okabayashi’s space**	Nerve-sparing procedures, RH, LUNA procedure, bowel resection for DIE or GC, RVH, SLNB	Ureter surgery for DIE or GC		Middle rectal vessels, PP, hypogastric nerves
**Fourth space**	Nerve-sparing procedures during DIE or GC	Ureter surgery for DIE or GC		Vesico-uterine ligament vessels
**Retropubic space**	Anterior exenteration, pelvic anterior peritonectomy, bladder endometriosis	MESH removals, ureteric re- implantation, retropubic TVT, anterior vaginal compartment repairs, Burch colposuspension, MMK procedure		Veins of Santorini, Dorsal vein of clitoris
**VU/VV space**	TLH, RH, RVH, nerve-sparing procedures, management of DIE, vaginal cuff resection	Vesico- uterine/vaginal fistula repair, bladder/ureter endometriosis, transvaginal cystocele operations, sacrocolpopexy	CS, Cesarean hysterectomy, LA/A cerclage, CS scar ectopic excision	uterine artery, superficial uterine vein, ureter branch of the uterine artery, superior vesical vein, cervicovesical vessels
**Rectovaginal space**	RH, rectovaginal fistula repair, treatment of pelvic adhesions, bowel resection for DIE	Sacrocolpopexy, uterosacral ligament suspension,		Vaginal, presacral veins, middle rectal vessels
**Presacral space**	bowel resection for DIE, presacral neurectomy, TME for GC, initiation of para-aortic lymphadenectomy	Sacrocolpopexy, hysterocolpopexy,		Common iliac, middle sacral vessels, inferior mesenteric artery, ureters, superior hypogastric plexus, hypogastric nerves

LPS—lateral paravesical space; MPS—medial paravesical space; VU/VV—vesicouterine/vesicovaginal; RH—radical hysterectomy; RVH—radical vaginal hysterectomy; DE—deep endometriosis; TLH—total laparoscopic hysterectomy, TME—total mesorectal excision; CS—cesarean section; GC—gynecological cancer; LA/A—laparoscopic/abdominal; CORM—corona mortis; MMK—Marshall–Marchetti–Krantz procedure; SLNB—sentinel lymph node biopsy.

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
