# Peer review of "Avascular Spaces of the Female Pelvis—Clinical Applications in Obstetrics and Gynecology"

_jcm, 2020, doi:10.3390/jcm9051460_

Round 1
Reviewer 1 Report
This is a very nice and detail description of female pelvic avascular spaces. I believe this will be a very informative article for all gynecologic surgeons.
- The surgical picture qualities needs to be improved if possible.
- In line 98, authors mention that PVS can be exposed by transecting round ligament. this is correct but I think they also need to mention that transecting round ligament even though may make it easier but it is not a necessary step.
- The paravaginal repair can also be performed by opening the lateral paravesical space down to urogenital diaphragm and identifying arcus tendineous ligament without opening medial paravesical space. If authors agree, please add that for completion of approaches.
- In line 116, authors state, " 9 and 11 o'clock opens left PVS " . I believe they mean right PVS. please describe side in relation to patient and not the surgeon.
- In line 117, Also, it will open the space lateral to vesicouterine ligament ( paper states medial). please correct if you are in agreement.
- In line 139, please clarify the Corona Mortis . Is this same as accessory obturator vessels or author feels corona mortis is different structure. Please mention accessory obturator vessels or clarify if they are same.
- In figure 5, It is marked as LS on the picture but written as LA in footnote. please correct if agree.
- In figure 5, MPS is written in footnote as medial para rectal space. This should be medial paravesical space. please correct it.
- In line 190, authors mention "to enter Latzko's space need to transect round ligament. Even though it is correct and possibly makes it easier but that is not necessary step and space can be dissected without transecting round ligament.
- In line 293, authors state " right avascular space" . I believe that may cause some confusion. It may be better to state " correct avasculare ".
- In Figure, Please mark the side based on patient orientation not surgeon orientation. sides are vice versa.
- In line 428; I believe authors describing approach from left side. if I am correct, I think it should say " pull the rectosigmoid to right ( not left) and make the incision to left of (not right of) sigmoid peritoneal attachment. Please clarify.
- excellent description of pararectal spaces. Authors describe, the area lateral to rectum and medial to uterosacral ligament as both anterior part of presacral space and also posterior part of rectovaginal space. The Okabayahi and Latzko space is described very well. The authors describe the Okabayashi's space as equivalent to medial para-rectal space. Their anatomic definition may be correct. However, leaves the definition of space between uterosacral ligaments and lateral portion of rectum between rectovaginal space and presacral space ambivalent. Could this space be defined as medial para-rectal space . The lateral para-rectal space be divided to two spaces ( Latzko's and Okabayashi's space) . Please make comments or describe the space better. In anterior resection, surgeon can dissect medial to uterosacral ligament and open the pararectal space.
- This is wonderful description of pelvic retro-peritoneal anatomy and avascular space. This is very informative paper for gynecologic surgeons.
- I congratulate the authors for wonderful manuscript.
Reviewer 2 Report
Interesting delineation of anatomical spaces.
Pictures are on cadavers. It be nice to add intra operative images. Examples of using those spaces in different gynecologic surgeries can also be further elaborated on to increase clinical significance.
Reviewer 3 Report
An exceptionally interesting and important topic was raised in a given publication. Avascular spaces of the female pelvis using a large number of figures have been described in great detail. It took a lot of work to prepare this publication, which makes it very valuable.
All retroperitoneal spaces in women that play an important role in obstetrics were described for the first time in this article. Accurate description and analysis of all these spaces is extremely important and can have a huge impact in both obstetrics, gynecology, surgery and urology.
The work is an extremely important position for many specializations, where it can be used.
